# Identification of Nanobodies against the Acute Myeloid Leukemia Marker CD33

**DOI:** 10.3390/ijms21010310

**Published:** 2020-01-02

**Authors:** Ema Romão, Ahmet Krasniqi, Laila Maes, Camille Vandenbrande, Yann G.-J. Sterckx, Benoit Stijlemans, Cécile Vincke, Nick Devoogdt, Serge Muyldermans

**Affiliations:** 1Laboratory of Cellular and Molecular Immunology, Vrije Universiteit Brussel, 1050 Brussels, Belgium; Ema.Estevens.Romao@vub.be (E.R.); Laila.Maes@vub.be (L.M.); Camille.Vandenbrande@vub.be (C.V.); bstijlem@gmail.com (B.S.); Cecile.Vincke@vub.be (C.V.); 2In Vivo Cellular and Molecular Imaging, Vrije Universiteit Brussel, 1090 Brussels, Belgium; Ahmet.Krasniqi@vub.be (A.K.); nick.devoogdt@vub.be (N.D.); 3Laboratory of Medical Biochemistry and the Infla-Med Centre of Excellence, University of Antwerp (UA), Campus Drie Eiken, Universiteitsplein 1, 2610 Wilrijk, Belgium; Yann.Sterckx@uantwerpen.be; 4Laboratory of Myeloid Cell Immunology, VIB, 1050 Brussels, Belgium

**Keywords:** nanobody, CD33, acute myeloid leukemia

## Abstract

Nanobodies (Nbs) are the smallest antigen-binding, single domain fragments derived from heavy-chain-only antibodies from Camelidae. Among the several advantages over conventional monoclonal antibodies, their small size (12–15 kDa) allows them to extravasate rapidly, to show improved tissue penetration, and to clear rapidly from blood, which are important characteristics for cancer imaging and targeted radiotherapy. Herein, we identified Nbs against CD33, a marker for acute myeloid leukemia (AML). A total of 12 Nbs were generated against recombinant CD33 protein, out of which six bound natively CD33 protein, expressed on the surface of acute myeloid leukemia THP-1 cells. The equilibrium dissociation constants (K_D_) of these six Nbs and CD33 range from 4 to 270 nM, and their melting temperature (Tm) varies between 52.67 and 67.80 °C. None of these Nbs showed leukemogenicity activity in vitro. The selected six candidates were radiolabeled with ^99m^Tc, and their biodistribution was evaluated in THP-1-tumor-bearing mice. The imaging results demonstrated the fast tumor-targeting capacity of the Nbs in vivo. Among the anti-CD33 Nbs, Nb_7 showed the highest tumor uptake (2.53 ± 0.69 % injected activity per gram (IA/g), with low background signal, except in the kidneys and bladder. Overall, Nb_7 exhibits the best characteristics to be used as an anti-CD33 targeting vehicle for future diagnostic or therapeutic applications.

## 1. Introduction

Acute myeloid leukemia is an aggressive clonal malignancy of myeloid progenitors characterized by clinical and biological heterogeneity [1]. It is also the most common leukemia in adults, accounting for 30% of all leukemias, with over 400,000 cases worldwide, in 2018 [2,3]. To date, human leukemia stem cells (LSCs) causing AML are the most and best-studied cancer-stem-cell population [4,5]. The current standard of care for medically fit patients consists of intensive chemotherapy, followed by allogeneic hematopoietic transplantation. Despite these aggressive treatments, there is high risk of relapse, with cure rates between 35% and 40% for younger, fitter patients, while the prognosis is worse for patients above 65 years of age, with a five-year survival below 5% [6,7]. The risk of relapse remains high for multifactorial reasons, mainly due to the LSC plasticity that resists the different selection pressures from chemotherapeutics [8,9].

Although multi-agent chemotherapy is the preferred treatment for AML, monoclonal antibodies (mAbs) have emerged as an attractive therapeutic option. Their ability to selectively target leukemia cells minimizes the systemic toxicity, resulting in an improved survival outcome. During the last decade, a lot of antibody-based therapy research against AML has been dedicated toward the identification and exploration of new antigen targets [10]. This yielded several monoclonal antibodies in clinical trials for the treatment of AML. Furthermore, encouraging results have been obtained by linking mAbs with potent chemicals, to form antibody–drug conjugates (ADCs) or with radionuclides to perform radioimmunotherapy [1,11].

One of the AML antigens that has been evaluated for the development of mAb-based therapies for AML is the sialic-acid-binding immunoglobulin-like lectin (siglec) CD33 protein. This 67 kDa protein is a myeloid differentiation antigen, with endocytic proprieties and a role in regulating leukocyte functions in inflammatory and immune responses [12,13]. The main rationale of using CD33 as an AML therapeutic target is its widespread overexpression across all subtypes of AMLs, with up to 90% of LSCs expressing CD33 [13,14]. Several anti-CD33 mAbs and ADCs have been developed and tested for the treatment of AML [6,10]. However, their limited efficacy and toxicities uncover the requirement for further development, focusing on the improvement of current therapeutic approaches, as well as on generating new alternatives.

We would like to propose nanobodies (Nbs) as a potential alternative to combat AML and LSCs in particular. An Nb is the smallest intact antigen-binding fragment of naturally occurring heavy-chain-only antibodies that has evolved to be fully functional in Camelidae. Their unique and well-characterized properties enable them to surpass conventional mAbs in terms of recognizing uncommon or hidden epitopes of protein targets, tailoring of plasma half-life, and drug format flexibility. In addition, Nbs are shown to be nonimmunogenic in humans and are easy to manufacture [15,16,17,18]. Furthermore, their favorable biochemical, biophysical, and pharmacological properties, combined with the simplicity of formatting them into multifunctional protein therapeutics, make them attractive targeting vehicles to be used for diagnosis and treatment of AMLs [19,20,21].

In the present work, we report the generation and characterization of Nbs targeting the AML marker CD33, in terms of their biophysical and in vivo targeting properties.

## 2. Results

### 2.1. Nanobodies Were Generated against Recombinant Human CD33

An Nb gene library, in pMECS phage-display vector [22], of 10^8^ transformants was constructed from peripheral blood of a llama immunized with recombinant CD33. After four rounds of phage-display selection on recombinant CD33 coated on microtiter plates, the periplasmic extracts of 78 individual clones were confirmed to bind the recombinant human CD33 ectodomain in an ELISA. After DNA sequencing, 12 different anti-CD33 Nbs were identified (Figure 1). These Nbs were divided into 10 different families, based on the amino acid sequence from the third complementarity determining region (CDR3). One of the Nb-families contained three representatives, namely Nb_7, Nb_21, and Nb_22, with up to five-point mutations. These three Nbs, belonging to the same family, are derived from a single B cell lineage (sharing the same *V-D-J* rearrangement). Nbs from the same family will recognize an identical epitope; however, the affinity for the antigen can be different due to somatic hypermutations that accumulated as a result of the affinity maturation process during immunization of the llama.

Analysis of the amino acid sequence of the Nbs revealed the presence Y (or H) at position 42, and R at position 50 (positions refer to the ImMunoGeneTics (IMGT) amino acid numbering). These hallmark amino acids indicate that the V-domain originated from a heavy-chain-only antibody [23]. The Nb_12 possesses V42 and P50, and these amino acids are encountered in VH domains of classical hetero-tetrameric antibodies [24,25]. However, the presence of E118 and D119 instead of W118 and G119 indicates that this Nb_12 would fail to interact with a VL domain; hence, it was also derived from a heavy-chain-only antibody [24].

All twelve Nbs were expressed, extracted from the periplasm, and purified by immobilized metal affinity chromatography (IMAC) and size exclusion chromatography (SEC). The SEC profile gave one single symmetrical peak (Figure 2A), reflecting the good solubility and homogeneity of the Nb. The separation of the proteins on sodium dodecyl sulphate polyacrylamide gel electrophoresis (SDS-PAGE) of each sample revealed one single band with a molecular weight around 15,000, as expected for the size of an Nb, after Coomassie staining and western blot (Figure 2B,C).

### 2.2. In Vitro Characterization of the Anti-CD33 Nanobodies

The ability of the generated anti-CD33 Nbs to recognize native human CD33 when expressed on the cell membrane was evaluated by flow cytometry, using THP-1 cells. Anti-CD33 Nbs were only considered as binders if the difference in mean fluorescence intensity (ΔMFI) signal was at least three times higher than the signal obtained with a non-targeting Nb. The ΔMFI obtained with anti-CD33 Nbs are shown in Figure 3A,B. From all generated anti-CD33 Nbs, six Nbs (Nb_7, Nb_12, Nb_16, Nb_21, Nb_22, and Nb_87) were shown to bind CD33 protein expressed on THP-1 cells.

The impact of the six Nbs recognizing CD33 on the proliferation of THP-1 cells was evaluated via the alamarBlue assay (Figure 3C). The alamarBlue signal correlates with the number of viable cells. The proliferative signal obtained after four hours of incubation of THP-1 cells, with 5 µg of each anti-CD33 Nb, was the same as the signal obtained from the non-targeting Nb, indicating that the proliferative capacity of THP-1 cells was not affected by the presence of anti-CD33 Nbs, under our experimental conditions.

The binding kinetics of selected anti-CD33 Nbs were determined by surface plasmon resonance (SPR), on immobilized biotinylated recombinant CD33 protein. The kinetic parameters are shown in Table 1, whereas a sensorgram example is shown in Figure 4A. The calculated kinetic rate association constant (k_a_) for all six Nbs was between 10^4^ and 10^7^ M^−1^·s^−1^, whereas the kinetic rate dissociation constant (k_d_) ranged from 2 × 10^−3^ to 10^−2^ s^−1^. The equilibrium dissociation constants (K_D_) were determined to be between 4 and 270 nM (Table 1).

The SPR instrument was also used for epitope binning. To evaluate the epitope-specificity of the selected anti-CD33 Nbs on the biotinylated recombinant human CD33 protein, we first saturated the Nb binding sites of one anti-CD33 Nb and then added a mixture of the same anti-CD33 Nb, with different anti-CD33 Nbs (Table 2 and Figure 4B,C). As expected, the epitope of Nb_7 was overlapping with that for Nb_21 and Nb_22, since all three Nbs share the same CDR3 sequences. It is well established that Nbs with homologous CDR3 sequence bind to the same epitope on the antigen [26]. Furthermore, the epitope of Nb_7 (and Nb_21 and Nb_22) was shown to overlap with that of Nb_87. Moreover, the obtained results showed three independent epitopes targeted by Nb_12, Nb_16, and Nb_7.

To examine the thermal stability of anti-CD33 Nbs, we evaluated the Tm by using the ThermoFluor^®^ assay (Table 1 and Figure 5). The Tm of a protein is a measure of its stability, which is a parameter considered to be an important indicator of good pharmaceutical properties. The Tm of anti-CD33 Nbs ranged from 52.67 ± 0.04 to 67.80 ± 0.08 °C. This is above the temperature of 50 °C, which is needed for practical labeling of anti-CD33 Nbs with ^99m^Tc for the in vivo characterization experiments.

### 2.3. Biodistribution of ^99m^Tc-Labeled Nbs

To evaluate the in vivo targeting potential of anti-CD33 Nbs, the CD33-specific Nbs and the non-targeting Nb were labeled with ^99m^Tc. Although Nb_22 had the highest thermostability (Tm 58 °C) of the three Nbs of the same family (Tm of Nb_7 = 54 °C, and Nb_21 = 53 °C), the Nb_22 failed to withstand the labeling conditions. Hence, it was not used for in vivo biodistribution studies. All successfully ^99m^Tc-labeled Nbs showed a radiochemical purity of >95%. Biodistribution and tumor targeting of ^99m^Tc-Nbs was assessed in mice bearing THP-1 tumors via micro-SPECT/CT imaging, at 1 h post-injection and necropsy study at 1.5 h post-injection. As illustrated in Figure 6, micro-single photon emission computed tomography/X-ray computed tomography (SPECT/CT) images of mice showed fast tumor targeting of anti-CD33 Nbs, with high accumulation of radioactivity in kidneys and bladder for all ^99m^Tc-labeled Nbs. The kidney and bladder signals are antigen-unrelated and due to the renal clearance route of radiolabeled Nbs, an observation that is also made for other small hydrophilic radioconjugates [27]. Mice injected with ^99m^Tc-Nb_21 showed higher accumulation of radioactivity in liver and spleen compared to the group of mice injected with the other ^99m^Tc-labeled Nbs. This could be due to cross reactivity of Nb_21 with mouse CD33, as mouse and human CD33 share 62% amino acid sequence identity for their ectodomain, but this needs further investigation. No significant difference in tumor uptake was observed in mice injected with ^99m^Tc-Nb_16 and ^99m^Tc-ctrl_Nb (the control Nb is an anti-idiotypic described by Lemaire et al. [28] that was selected for its low background in naive mice).

The ex vivo biodistribution study confirmed the significant higher tumor targeting for anti-CD33 Nbs (tumor uptake from 1.91 ± 0.64%IA/g to 2.53 ± 0.69%IA/g), compared to Nb_16 (tumor uptake of 0.70 ± 0.13%IA/g) and to the non-targeting control ^99m^Tc-ctrl_Nb (tumor uptake of 0.16 ± 0.04%IA/g) (Figure 7A,B and Table 3). Among all anti-CD33 Nbs, Nb_7 showed the highest tumor uptake (2.53 ± 0.69%IA/g), with low background signal, except in kidneys and bladder.

## 3. Discussion

CD33 is a prominent biomarker in AML, a devastating disease. Although the 12-month survival of AML patients has increased up to 30% over the last three decades, the five-year survival remains unsatisfactory (below 5%) [6,29]. To improve these numbers, antibody-based therapies have been pursued since the late 1980s. However, they all showed very limited success due to low efficacy, toxicity, and immunogenicity issues [10].

Despite this, the clinical development of novel antibody-based therapies remains an interesting route to generate innovative therapeutics for AML. We would like to propose Nbs, a single-domain antigen-binding fragment, as an alternative to further extend the antibody-based arsenal against AML. Nbs have unique and beneficial biophysical, biochemical, and pharmacological proprieties that confer them the ability to rival and surpass conventional therapeutic antibodies. Indeed, the successful development of Nbs has already been reported against tumor-associated cell-surface markers such as HER2, CD20, PD-L1, CD47, CEA, and PSMA [20,30,31,32,33,34].

In this study, we retrieved 12 Nbs against CD33, an AML marker. Among these 12 Nbs, three Nbs (Nb_7, Nb_21, and Nb_22) showed significant homology in their CDR3 region, and minor amino acid variations in the remaining sequence. These clones are derived from the same *V-D-J* recombination event, revealing their common B-cell origin. Although the somatic hypermutation (or PCR errors) might have affected the affinity for the antigen, the dominant importance of the CDR3 in recognizing the epitope means they will bind to the same epitope [26], a finding that was confirmed from our epitope binning, using SPR.

After expression and purification, the 12 candidates were assessed in an in vitro binding to human CD33, natively expressed on the cell surface of an AML cell line (THP-1). Only six out of the 12 Nbs scored positive in recognizing the THP-1 cells, and these Nbs were taken for further characterization of affinity, thermostability and cellular effects.

To determine the in vitro therapeutic potential of the selected anti-CD33 Nbs, we evaluated the impact of anti-CD33 Nbs on the proliferation of THP-1 cells by performing the alamarBlue assay. This was explored based on the increasing body of evidence that CD33, upon activation, negatively regulates inflammatory and immune responses through inhibitory effects on signaling cascades. Moreover, these tyrosine kinase-driven pathways can result in reduced cellular activation and proliferation [35]. However, our results from the alamarBlue experiments revealed that none of the anti-CD33 Nbs affected the proliferative capacity of the THP-1 cells. This may be due to a limitation of the assay or the conditions we have used. Likewise, we have not yet assessed the possible antagonistic potential of our anti-CD33 Nbs. Nevertheless, based on our current data, it seems that our anti-CD33 Nbs will need some additional engineering in order to gain therapeutic potential.

Since two completely independent epitopes were recognized by Nb_12 and Nb_16, while Nb_7 (and its homologues Nb_21 and Nb_22) and Nb_87 targeted an overlapping epitope, it is possible to generate biparatopic anti-CD33 constructs with potentially improved affinities (avidity), specificities, and functionalities [36]. If explored in a multivalent manner, the generated Nbs could make use of CD33′s endocytic properties and enhance their own delivery of toxic-loads potential and/or trigger the mentioned signaling cascades [10,35,37].

Furthermore, the anti-CD33 Nbs could be used in a drug conjugate modality. Hereby, anti-CD33 Nbs can be developed as targeting vehicles to specifically deliver toxins to the target cancer cells, in the so-called antibody–drug conjugates (ADCs). A similar approach has also been evaluated by using classical anti-CD33 antibodies as targeting vehicles. One example of an ADC agent for the treatment of AML is gentuzumab ozogamicin (Mylotarg, GO, Pfizer), an IgG4 anti-CD33 antibody conjugated to calicheamicin (a chemotherapeutic DNA cleaving agent). In 2010, GO was withdrawn from the market due to clinical indications of early death, added toxicity, and limited clinical benefits [38]. Since GO, seven other drugs for the treatment of AML were approved by the Food and Drug Administration (FDA), over the last two years; however, treatment outcomes remain unsatisfactory for many patients [10].

Nbs can also be used in targeted radionuclide therapy (TRNT), whereby the Nb acts as vehicle to direct a therapeutic radionuclide to the tumor, as shown in mouse models of lymphoma, multiple myeloma, breast, and ovarian cancer [20,21,28,39]. A similar approach could also be pursued by using anti-CD33 Nbs. However, kidneys will be the main dose-limiting organ. Indeed, the rapid renal clearance of the unbound labeled Nb-fraction will result in a decreased radiation of other non-target organs, but it may lead to radiation-induced nephropathy. Different strategies can be assessed, to reduce the renal accumulation of radiolabeled Nbs, such as the removal of the His_6_-tag or co-injection of positively-charged amino-acids or Gelofusin [21,40].

Anti-CD33 Nbs could also be incorporated into more complex constructs to combat AML. A new approach for therapeutics in AML has been focusing on chimeric antigen receptor (CAR) T cells. In this approach, patient-derived autologous T cells are transduced with an extracellular anti-CD33 antibody fragment (either single chain fragment variable or Nb) that is genetically fused with intracellular T-cell activation domains [41,42]. This strategy not only directs T cells to AML; it also activates them upon recognition [41]. It constitutes an alternative, promising anti-CD33 immunotherapeutic approach that further builds on the clinical approval of anti-CD19 CAR T cells (Kymriah) to treat B-cell acute lymphoblastic leukemia and non-Hodgkin lymphoma [43]. Alternatively, the generated anti-CD33 Nbs could easily be employed to construct a Bi-specific T-cell engagers (BiTE)-like format, so that CD33 expressing AML blasts become engaged with T cells. Instead of using traditional single-chain variable fragments (scFv), our Nb against CD33 will be linked to a binder of T-cell receptors (e.g., CD3), yielding a construct with a smaller size and higher modularity, provoking little, if any, stability and solubility issues.

Due to their small size, Nbs are rapidly cleared from the body and penetrate better into the target tissues compared to the conventional antibodies [44,45]. Based on these characteristics, Nbs targeting various tumors or immune biomarkers haven been generated and developed as imaging agents in different mouse models of cancer. An anti-HER2 Nb has also been evaluated in a phase I clinical positron emission tomography (PET) trial, showing specific targeting of HER2-positive tumor lesions in patients [46]. Phase II clinical trials with anti-HER2 Nb as an imaging agent are currently ongoing (NCT03924466 and NCT03331601). In this study, we evaluated the in vivo biodistribution profile of ^99m^Tc-radiolabeled anti-CD33 Nbs in mice bearing THP-1 tumors. SPECT/CT imaging at 1 h post-injection showed high nonspecific accumulation in the kidney and bladder for all ^99m^Tc-radiolabeled Nbs. The high kidney and bladder accumulation of small hydrophilic compounds is a well-known phenomenon, which occurs as a result of a rapid renal clearance of the unbound fraction [47,48]. In addition, a rapid tumor targeting was observed for ^99m^Tc-labeled Nb_12, Nb_87, and Nb_7, with low background signal, except in the kidneys and bladder. ^99m^Tc-Nb_21 also showed high tumor accumulation but also a high accumulation in liver and spleen, potentially due to the cross-reactivity with mouse CD33. The non-targeting control Nb showed no tumor accumulation, confirming the specific targeting of anti-CD33 Nbs. Surprisingly, ^99m^Tc-Nb_16 showed very low tumor accumulation, comparably to that of non-targeting ^99m^Tc-ctrl_Nb. The ex vivo biodistribution results confirmed the in vivo results, with a significant higher tumor targeting for anti-CD33 Nb_12, Nb_87, Nb_21, and Nb_7 compared to Nb_16 and to the non-targeting control ^99m^Tc-ctrl_Nb. However, when comparing all anti-CD33 Nbs, Nb_7 showed the highest tumor uptake (2.53 ± 0.69%IA/g), with the lowest background signal.

These results show the diagnostic potential of anti-CD33 Nbs to be used as imaging agents for diagnosis of AML and/or to follow up the treatment response in treated patients. Indeed, CD33 is expressed in 90% of all AML cases, being displayed on at least one subset of the leukemic blasts and with increasing evidence suggesting its expression on LSCs [10,49,50]. This wide surface expression makes CD33 a very interesting diagnostic target. Currently, the diagnosis and prognosis of AML is performed after an invasive methodology via single-point bone-marrow biopsies, followed by cytogenetics [51], and not via a whole-body representative manner. Hence, the development of an Nb-based noninvasive imaging agent in the diagnosis and prognosis of AML remains an open opportunity. Madabushi et al. [51] showed the anti-CD33 PET imaging using [^64^Cu]-DOTA labeled anti-CD33 antibody for the detection of AML with high sensitivity and specificity. In this context, Nbs can also be considered as promising imaging agents due to their fast tumor targeting and rapid blood clearance of the unbound fraction, resulting in high tumor-to-background ratios early after injection.

## 4. Material and Methods

### 4.1. Anti-CD33 Nanobodies Generation, Selection, and Expression

Nbs against recombinant human CD33 were generated as previously described [22]. Briefly, a llama (*Lama glama,* housed in Lamasté, Leopoldsburg, Belgium) was immunized six times, at weekly intervals, with 150 µg of recombinant human CD33 ectodomain (Acro Biosystems, CD3-H5226, Asp 18–His 259) supplemented with Gerbu LQ 3000 adjuvant (Gerbu Biotechnik GmbH). Four days after the last immunization, lymphocytes were extracted from the anti-coagulated blood of the llama, and cDNA was prepared and used as a template in a two-step, nested PCR that amplified the gene fragments encoding the variable domain of the heavy-chain-only antibodies. The amplified DNA fragments encoding Nbs were ligated into a pMECS phagemid vector and transformed into *Escherichia coli* TG1 cells (Lucigen), resulting into a cloned Nb-library of 10^8^ transformants. After four rounds of phage selection, 78 individual clones were cultured, and their periplasmic proteins were tested for specificity against recombinant human CD33 protein in an ELISA. The ELISA-positive clones were selected for DNA sequencing (VIB Genetic Service Facility, Antwerp, Belgium) of the cloned Nbs. The inserted DNA sequences were translated into amino acid sequences, analyzed using the CLC Main Work Bench 8 software (Qiagen, Hilden, Germany), aligned according to the international ImMunoGeneTics database (http://imgt.cines.fr, accessed on 25 October 2019), and manually annotated [25].

Recombinant pMECS plasmids encoding for HA- and His_6_-tagged Nbs were transformed in *Escherichia coli* WK6 cells for the periplasmic expression of Nbs. Following extraction of the Nb proteins from the periplasm, they were purified by IMAC and SEC, as previously described [22]. The purity of the Nbs was determined by SDS-PAGE and Coomassie blue staining. A Western blot developed using a mouse anti-HA tag monoclonal antibody (BioLegend, San Diego, CA, USA) and goat anti-mouse IgG-HRP (Sigma-Aldrich, Saint Louis, MO, USA) conjugated antibody was performed to confirm the purity and identity. For in vivo imaging studies, Nbs were re-cloned in the plasmid pHEN6c, containing only the carboxyterminal His_6_ tag, and were expressed and purified, as described above. Besides anti-CD33 Nbs, a non-targeting control Nbs was used for in vitro binding and specificity experiments. This was Nb-ER19 from Moonens et al. [52], an Nb directed against a bacterial adhesion protein of *Helicobater pylori* (BabA), which is referred to herein as αBabA Nb_19.

### 4.2. THP-1 Maintenance and Preparation

Human acute myeloid leukemia THP-1 cells (ATCC, Manassas, USA) were grown as a suspension culture and maintained at 37 °C and 5% CO_2_ in RPMI-1640 media, supplemented with 10% fetal bovine serum (Thermo Scientific), 100 U/mL of penicillin-streptomycin (Sigma-Aldrich, Zwijndrecht, Belgium), and 0.05 mM of β-mercapto-ethanol (Sigma-Aldrich, Zwijndrecht, Belgium). Before each experiment, cell viability was checked by using trypan-blue exclusion.

### 4.3. Cell-Binding Assay

In vitro targeting of anti-CD33 Nbs was evaluated on THP-1 cells, using flow cytometry. THP-1 cells (2 × 10^5^/100 µL) were preincubated, with or without 1 µg of each anti-CD33 Nb and the non-targeting control Nb, for 1 h at 4 °C. After washing cells with cold PBS, cells were incubated with 150 ng of FITC-anti-HA.11 Ab (BioLegend, San Diego, CA, USA), at 4 °C, for 30 min. In another condition, THP-1 cells incubated only with 200 ng of APC anti-human CD33 antibody (BioLegend) were used as a positive control. Cells incubated with 200 ng of APC anti-Mouse IgG1 antibody κ (BioLegend) were used for the isotype control. Flow cytometry was performed on a FACS CANTO II analyzer, and data were processed by using FlowJo Software (BD Biosciences, San Jose, CA, USA).

### 4.4. Leukemogenic Activity Assay

To evaluate the in vitro cytotoxicity potential of anti-CD33 Nbs on THP-1 cells, we used the alamarBlue assay (Biorad). In a 96-well plate, 12 500 THP-1 cells were plated per well and treated with 5 µg of each anti-CD33 Nb or the non-targeting Nb. Untreated wells were also included for the evaluation of the background cytotoxicity activity. The wells were supplemented with 1/10 volume of a 10× alamarBlue reagent and incubated for 48 h. Absorbances at 570 and 600 nm were measured every 12 h, via VersaMax ELISA Microplate Reader (Molecular Devices), using the SoftMax^®^ Pro software (Molecular Devices, San Jose, CA, USA).

### 4.5. Surface Plasmon Resonance—Affinity Determination and Epitope Binning

The affinity of the generated anti-CD33 Nbs was determined by surface plasmon resonance, using the BIACORE-T200 (GE Healthcare, Freiburg, Germany). The biotinylated CD33 recombinant protein (Acro Biosystems, CD3-H82E7, Asp 18–His 259) was coupled to a streptavidin (SA)-coated sensor chip and used to evaluate the binding of a serial dilution series of the anti-CD33 Nbs. The SPR measurements were performed at 25 °C with HBS (20 mM of HEPES pH 7.4, 150 mM of NaCl, 0.005% Tween-20, and 3.4 mM of EDTA) as running buffer. The Nbs were injected sequentially in 2-fold serial dilutions, from 250 to 2 nM, at a flow rate of 10 µL·min^-1^. The association step was 300 s, the dissociation step was 400 s, and a two-step regeneration of 35 s at 30 µL·min^-1^, using 100 mM Glycine at pH 3.0, was performed. After the dissociation step, there was an additional stabilization step of 300 s. The rate kinetic constants were determined by a mathematical fitting of a 1–1 binding model using the BIACORE Evaluation software (GE Healthcare, Freiburg, Germany), and the k_df_/k_a_ ratio was used to determine the equilibrium dissociation constant (K_D_).

For the epitope binning experiments, an excess of Nb “A” was first injected for 300 s, at a concentration of 100 times of its K_D_ value, to saturate all epitopes available on the CD33 protein. This was followed by a second injection under the same conditions, but which consisted of a mixture of excess Nb “A” and Nb “B” (both at a concentration of 100 times of their K_D_ value). The RU was then monitored for 600 s, followed by a regeneration step, using 2 injections for 45 s each, at 30 µL·min^−1^, using 100 mM of Glycine at pH 3.0. All Nb-pairs were evaluated in all possible combinations.

### 4.6. Thermal Stability

The Tm of the anti-CD33 Nbs was determined via a ThermoFluor^®^ assay, using CFX Connect^TM^ Real-Time PCR (Biorad, Pleasanton, CA, USA) and the fluoroprobe Sypro^®^ Orange dye (Molecular Probes, Oregon, OR, USA), as previously described [53]. For the thermal denaturation, the samples were heated from 10 to 95 °C, with stepwise increments of 0.5 °C per 30 s, and a 10 s hold step at every point, followed by the fluorescence reading. The Tm was calculated by using the Boltzmann equation.

### 4.7. Preparation of ^99mTc^ Labeled Nbs

Anti-CD33 and non-targeting control Nbs were radiolabeled with Technetium-99m (^99m^Tc) for the biodistribution study. Nbs were labeled with [^99m^Tc(H_2_O)_3_(CO)_3_]^+^ at their His_6_-tag via tricarbonyl chemistry, as previously described [54]. The ^99m^Tc-labeled Nbs were purified from the unbound [^99m^Tc(H_2_O)_3_(CO)_3_]^+^ via NAP-5 SEC (Sephadex, GE Healthcare, Machelen, Belgium), and filtered through a 0.22 µm filter (Millex, Millipore, Haren, Belgium). The radiochemical purity of radiolabeled Nbs was evaluated by instant thin layer chromatography (iTLC-SG, Pall Corporation, Hoegaarden, Belgium).

### 4.8. Animal Model

For biodistribution experiments, female CB17 SCID mice were subcutaneously inoculated in the right hind limb with 2 × 10^6^ THP-1 cells, under 2.5% isoflurane anesthesia (Abbott, Ottignies-Louvain-la-Neuve, Belgium). Tumors reached a maximal size of 200 ± 50 mm^3^. Prior to imaging, mice were anesthetized with 18.75 mg/kg of ketamine hydrochloride (Ketamine 1000 Ceva^®^, Ceva, Brussels, Belgium) and 0.5 mg/kg of medetomidine hydrochloride (Domitor^®^, Pfizer, Brussels, Belgium). Animal study protocols were approved by the Ethical Committee for Animal Experiments of the Vrije Universiteit Brussel (Project Identification code ECD: 18-272-5, date of approval 23 March, 2018).

### 4.9. In Vivo Biodistribution of Radiolabeled Nbs

Mice bearing subcutaneous THP-1 tumors were intravenously injected with ^99m^Tc-labeled anti-CD33 Nbs or a non-targeting ^99m^Tc-labeled control Nb (^99m^Tc-ctrl_Nb) (24.41–60.31 MBq; *n* = 3). For the in vivo binding studies, we preferred to use the anti-mouse idiotype Nb_R3B23, characterized by Lemaire et al. [28] as non-targeting control (here denoted as ctrl_Nb). At 1 h post-injection, micro-SPECT/CT imaging was performed, using the MILabs VECTor/CT. The CT-scan was set to 55 kV and 615 μA, resolution of 80 μm. The total body scan duration was 108 s. SPECT images were generated by using a rat SPECT-collimator (1.5 mm pinholes) in spiral mode, with 20 positions for whole-body imaging, 90 s per position. Images were reconstructed with 0.4 mm voxels with 2 subsets and 7 iterations, without post-reconstruction filter [20]. Images were analyzed by using Medical Image Data Examiner (AMIDE) software. Maximum intensity projections were generated using OsiriX Lite software (Pixmeo SARL, CH1233 Bernex, Switzerland).

After imaging, mice were killed for an ex vivo biodistribution study. Different organs, tissues, and tumors were isolated, and the present radioactivity was measured against a standard of known activity, using a γ-counter (Cobra II Inspector 5003, Canberra Packard), and expressed as %IA/g tissue, corrected for decay.

### 4.10. Statistical Analysis

Statistical analyses were conducted by using the one-way ANOVA, with Bonferroni’s multiple comparison test. The statistical difference in the figures is indicated as follows: * (*p* < 0.05), ** (*p* < 0.01), and *** (*p* < 0.001).

## 5. Conclusions

In this study, we showed the successful generation of different stable, high-affinity anti-CD33 Nbs. After an extensive in vitro characterization, anti-CD33 Nbs were radiolabeled with ^99m^Tc, and their biodistribution profile was evaluated in mice bearing tumors. Four anti-CD33 Nbs showed fast and specific in vivo tumor targeting, with a rapid renal clearance of the excess labeled material. Although our anti-CD33 Nbs can be used for in vivo imaging to diagnose or follow-up patients, the Nbs lack an inherent therapeutic activity. To generate a therapeutic outcome for AML, it will be necessary to engineer the anti-CD33 Nbs into BiTE or CAR-T cell constructs, or to decorate them with toxic drugs into ADC or with radionuclides for TRNT.

## Figures and Tables

**Figure 1 ijms-21-00310-f001:**
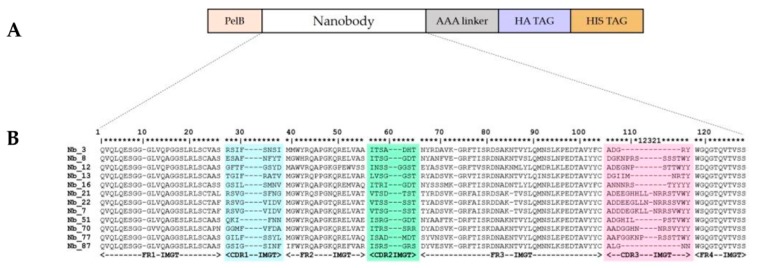
(**A**) Schematic representation of an Nb in the phagemid vector pMECS. Downstream of the PelB secretion sequence, the Nb-sequence is followed by a triple alanine linker, a hemagglutinin (HA), and hexa-histidine (His) tags. (**B**) Amino acid sequences of the anti-CD33 Nbs (numbering according to IMGT) [25]. The CDR1, CDR2, and CDR3 regions are highlighted in cyan, green, and pink, respectively. The amino acid sequence of the CDR3 region is displayed in alphabetical order. Position 10 in the framework region-1 (FR1-IMGT) and position 73 in the FR3-IMGT are gaps introduced to align to other V-GENE groups or subgroups. For Nb_7, Nb_21, and Nb_22, an amino acid deletion relative to other sequences occurred at position 85 (represented by a dash).

**Figure 2 ijms-21-00310-f002:**
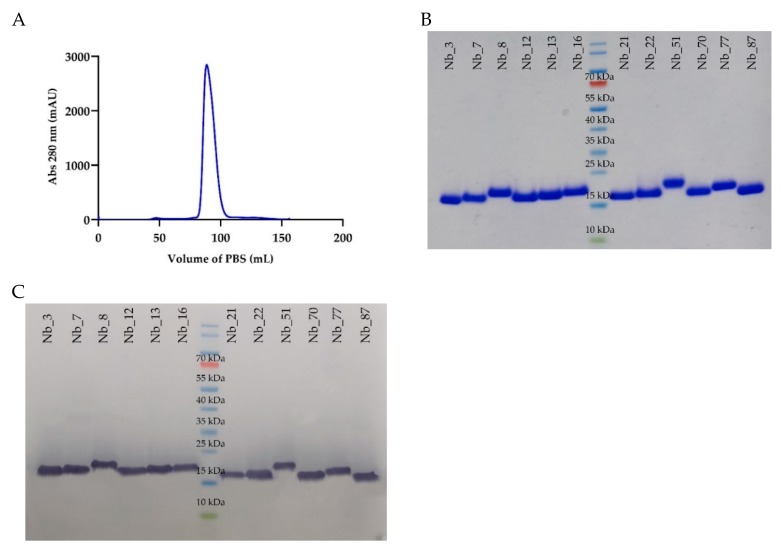
Purity of the anti-CD33 Nb preparations. (**A**) SEC profile of the Nb_12, showing a single peak of protein. (All other Nbs gave comparable chromatograms). (**B**,**C**) SDS-PAGE under reducing conditions, where proteins are revealed after staining with Coomassie blue (**B**) or by western blot, using a mouse anti-HA tag monoclonal antibody and a goat anti-mouse IgG horse radish peroxidase (HRP)-conjugated antibody (**C**). For both staining conditions and for each Nb preparation, only one single band was revealed.

**Figure 3 ijms-21-00310-f003:**
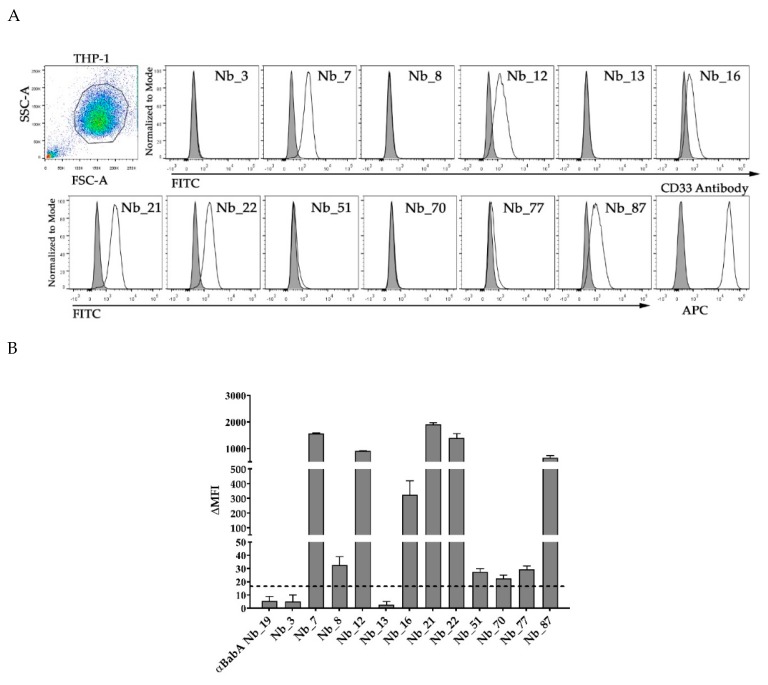
Six anti-CD33 Nbs bind native CD33 protein expressed on THP-1 cells, without affecting the cells’ in vitro proliferative capacity. (**A**) Individual histogram plots of flow cytometry analysis from the selected Nbs (clear peak) versus a non-targeting Nb (tinted peak). An anti-CD33 monoclonal antibody was used as positive control (clear peak), with an isotype-matched antibody as negative control (tinted peak). (**B**) Graphical representation of the ΔMFI values for the generated Nbs. The ΔMFI is defined as the MFI signal from THP-1 cells treated with Nb and HA-labeled monoclonal antibody subtracted with the MFI signal from cells and labeled monoclonal, but without Nb. An Nb was selected as a binder if its ΔMFI signal was at least three times higher than the one obtained with non-targeting binder (αBabA Nb_19), which defined the threshold (dashed line). (**C**) The impact of ant-CD33 Nbs on the proliferation of THP-1 cells was determined by the alamarBlue assay, in which the proliferative status measured by absorbance is translated into a bar plot. THP-1 cells were incubated for 48 h, with 5 µg of the selected anti-CD33 Nbs (gray bars) or a non-targeting Nb (αBabA Nb_19; light grey bar), or they were left untreated (patterned bar). Medium alone was also included as extra control condition (black bar).

**Figure 4 ijms-21-00310-f004:**
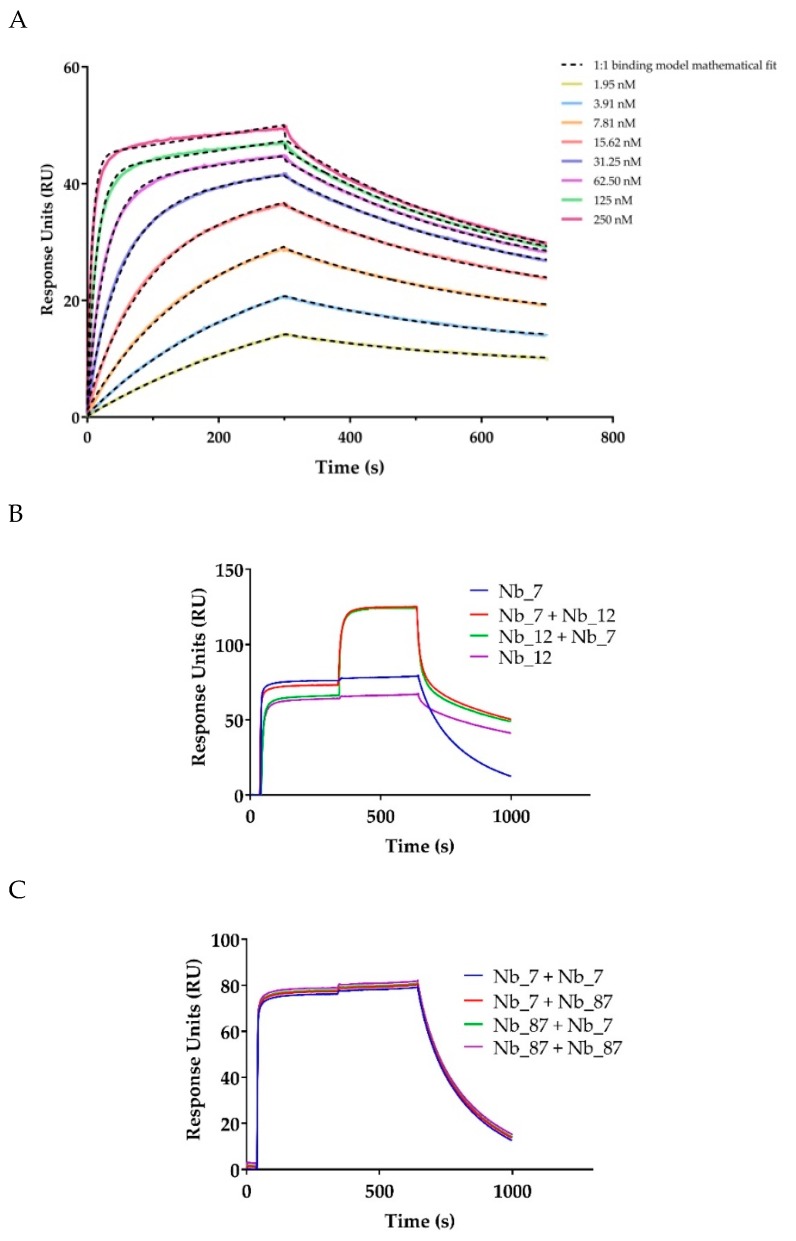
SPR sensorgram of Nb_12 on the CD33 ectodomain and epitope binning by SPR: (**A**) sensorgram of different concentrations (as indicated in the graph) of anti-CD33 Nb_12 binding to biotinylated recombinant human CD33 ectodomain. Kinetics were measured with a two-fold dilution series of Nbs (250–1.95 nM). The fitting of the binding curves using the 1:1 binding mathematical model calculated a K_D_ of 3.9 nM. (**B**,**C**) Examples of a noncompetitive (independent) and a competitive (overlapping) epitope binding of two Nbs for the same antigen, respectively.

**Figure 5 ijms-21-00310-f005:**
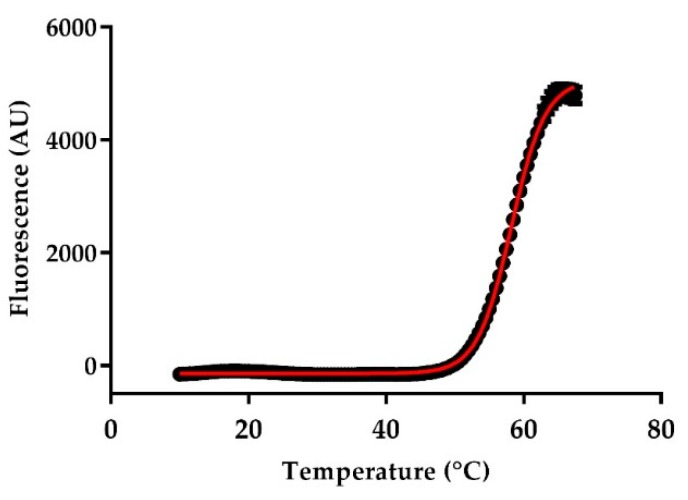
An example of Tm measurement of anti-CD33 Nb_12, as determined by ThermoFluor^®^ assay. The samples were heated from 10 to 95 °C, with stepwise increments of 0.5 °C per 30 s. AU stands for arbitrary units. The Tm for this Nb (58.00 ± 0.23 °C) was determined by using Boltzmann’s Equation.

**Figure 6 ijms-21-00310-f006:**
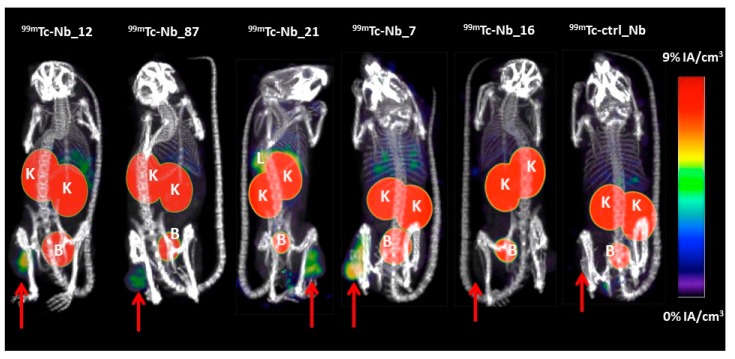
Biodistribution of ^99m^Tc-labeled anti-CD33 Nbs, and the non-targeting ^99m^Tc-ctrl_Nb in mice bearing THP-1 tumors. Micro-SPECT/CT images were obtained 1 h after intravenous injection of ^99m^Tc-labeled Nbs. Arrows indicate THP-1 tumors. K: kidney; B: bladder; and L: liver.

**Figure 7 ijms-21-00310-f007:**
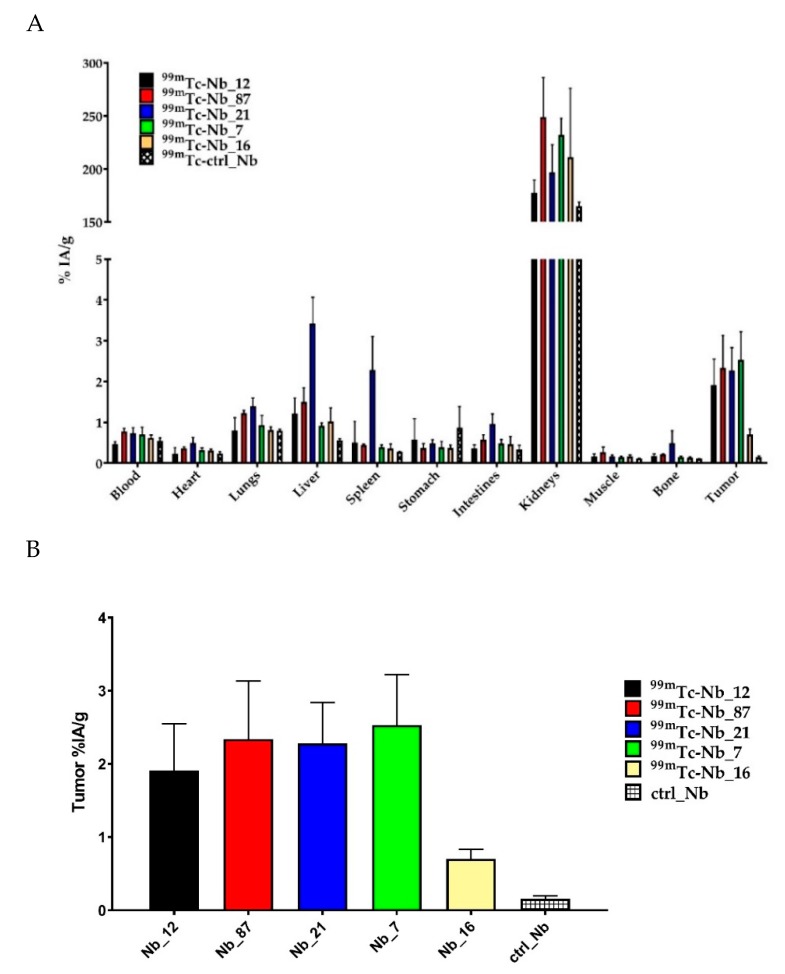
Graphical presentation of the biodistribution of the ^99m^Tc-labeled anti-CD33 Nbs and non-targeting control Nb (ctrl_Nb) in mice bearing subcutaneous THP-1 tumors (*n* = 3). (**A**) Ex vivo biodistribution results obtained 1.5 h after injection. Results are presented as mean %IA/g ± standard deviation. (**B**) Mice injected with ^99m^Tc-labeled Nb_12, Nb_87, Nb_21, and Nb_7 showed significant higher tumor uptake compared to the group of mice injected with non-targeting ^99m^Tc labeled ctrl_Nb.

**Table 1 ijms-21-00310-t001:** Kinetic association (k_a_), dissociation (k_d_), and equilibrium (K_D_) constants of the six anti-CD33 Nbs for CD33 ectodomain measured by SPR. The Tm value representing the thermal stability of the six Nbs was measured with the ThermoFluor^®^ method.

Nanobody	k_a_ (M^−1^·s^−1^)	k_d_ (s^−1^)	K_D_ (M)	Tm (°C)
Nb_7	1.1 × 10^6^	6.1 × 10^−3^	5.5 × 10^−9^	54.47 ± 0.05
Nb_12	5.4 × 10^5^	2.1 × 10^−3^	3.9 × 10^−9^	58.00 ± 0.23
Nb_16	1.3 × 10^4^	3.4 × 10^−3^	2.6 × 10^−7^	57.80 ± 0.03
Nb_21	1.3 × 10^6^	9.1 × 10^−3^	7.0 × 10^−9^	52.67 ± 0.04
Nb_22	1.5 × 10^6^	1.9 × 10^−3^	1.3 × 10^−8^	58.22 ± 0.03
Nb_87	9.3 × 10^6^	7.7 × 10^−3^	8.1 × 10^−8^	67.80 ± 0.08
αBabA Nb_19 *	2.8 × 10^6^	1.2 × 10^−3^	0.4 × 10^−9^	61.75 ± 0.14

* The binding parameters for αBabA Nb_19 (used as control Nb in in vitro experiments) are for the interaction with its cognate antigen BabA (*H. pylori*’s adhesion protein).

**Table 2 ijms-21-00310-t002:** Overview of the epitope binning experiments. Each pair of anti-CD33 Nbs is classified as noncompetitive if it recognizes a non-overlapping epitope, or as competitive if it binds to an overlapping epitope.

Nanobody	Nb_7	Nb_12	Nb_16	Nb_21	Nb_22	Nb_87
Nb_7	competitive	non-competitive	non-competitive	competitive	competitive	competitive
Nb_12	competitive	competitive	non-competitive	non-competitive	non-competitive	non-competitive
Nb_16	competitive	competitive	competitive	non-competitive	non-competitive	non-competitive
Nb_21	competitive	competitive	competitive	competitive	competitive	competitive
Nb_22	competitive	competitive	competitive	competitive	competitive	competitive
Nb_87	competitive	competitive	competitive	competitive	competitive	competitive

**Table 3 ijms-21-00310-t003:** Ex-vivo biodistribution of ^99m^Tc-Nbs in mice bearing THP-1 tumors.

	Nb_7	Nb_12	Nb_16	Nb_21	Nb_87	ctrl_Nb
Blood	0.70 ± 0.18	0.46 ± 0.06	0.62 ± 0.07	0.73 ± 0.14	0.78 ± 0.07	0.57 ± 0.09
Heart	0.32 ± 0.05	0.23 ± 0.15	0.31 ± 0.04	0.50 ± 0.13	0.37 ± 0.03	0.25 ± 0.06
Lungs	0.94 ± 0.24	0.80 ± 0.31	0.81 ± 0.08	1.40 ± 0.20	1.23 ± 0.07	0.78 ± 0.04
Liver	0.92 ± 0.07	1.22 ± 0.38	1.02 ± 0.34	3.42 ± 0.65	1.50 ± 0.35	0.57 ± 0.05
Spleen	0.39 ± 0.06	0.50 ± 0.53	0.36 ± 0.11	2.28 ± 0.82	0.44 ± 0.02	0.28 ± 0.01
Stomach	0.39 ± 0.15	0.57 ± 0.52	0.38 ± 0.07	0.48 ± 0.10	0.37 ± 0.11	1.02 ± 0.63
Intestines	0.48 ± 0.10	0.36 ± 0.09	0.46 ± 0.19	0.97 ± 0.24	0.58 ± 0.12	0.37 ± 0.13
Kidneys	232.04 ± 15.83	177.47 ± 12.14	211.16 ± 64.88	197.04 ± 25.96	249.11 ± 37.07	163.68 ± 4.78
Muscle	0.15 ± 0.03	0.16 ± 0.06	0.16 ± 0.04	0.16 ± 0.04	0.27 ± 0.13	0.12 ± 0.01
Bone	0.15 ± 0.03	0.17 ± 0.05	0.13 ± 0.02	0.49 ± 0.31	0.22 ± 0.00	0.11 ± 0.00
Tumor	2.53 ± 0.69	1.91 ± 0.64	0.70 ± 0.13	2.28 ± 0.56	2.34 ± 0.79	0.16 ± 0.04
Tumor/Blood	3.66 ± 0.94	4.15 ± 0.69	1.12 ± 0.09	3.11 ± 0.46	3.02 ± 1.08	0.27 ± 0.03
Tumor/Muscle	17.57 ± 5.13	12.56 ± 4.43	4.39 ± 0.88	14.03 ± 3.08	9.20 ± 3.51	1.31 ± 0.28
Tumor/Bone	16.37 ± 1.95	11.42 ± 3.60	5.36 ± 1.12	5.35 ± 1.66	10.64 ± 3.68	1.46 ± 0.43

Values are expressed as %IA/g and presented as mean ± standard deviation.

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
