# Peer review of "Identification of Nanobodies against the Acute Myeloid Leukemia Marker CD33"

_ijms, 2020, doi:10.3390/ijms21010310_

Round 1
Reviewer 1 Report
It is a very well written original paper regarding identification of a lead Nanobody against the acute myeloid leukemia marker CD33.
Please discuss further the future therapeutic applications of this Nanobody.
Author Response
Response to the reviewers
Manuscript ID: ijms-642026:
Identification of a lead Nanobody against the acute myeloid leukemia marker CD33
First of all, we would like to thank the reviewers for their positive and constructive feedback and helpful suggestions to improve our paper. The original sentences have been rephrased throughout the text to increase readability, to correct typo’s and/or remove inconsistencies in abbreviations. We amended the text of our manuscript to address all the comments raised by the reviewers in the following manner:
Reviewer #1
It is a very well written original paper regarding identification of a lead Nanobody against the acute myeloid leukemia marker CD33.
Please discuss further the future therapeutic applications of this Nanobody.
Answer from authors to comment: The discussion has been rewritten to remove major repetitions form the result section. In addition, paragraphs have been added to explain the possibilities to apply our anti-CD33 Nbs in therapeutic and non-invasive imaging Nb-based constructs. We also included references to what others have been doing in this respect.
Suggestion by reviewer:
English language and style: English language and style are fine/minor spell check required
Answer from authors to suggestion: The text has been revised to remove typo’s.
Reviewer 2 Report
The authors have given a reasonable account of the successful generation of CD33 Nanobodies and demonstration of their specificity. This will be of interest for those in the leukaemia field, seeking to develop new therapeutic agents. The authors must improve both the readability and presentation of their manuscript by addressing the comments below:
Title: there is no lead Nanobody candidate here. No experimental data presented gives clear data on a ‘lead’ candidate therefore remove from Title and abstract. inconsistence nomenclature of Nbs: e.g. sometimes Nb7 sometimes Nb 7. Check all Figures and legends for consistent usage line 22: tumour-bearing line 32: 437 000 – number looks weird, check other numbers too eg Line 114 line 34: a fruit of labor?? Line 52: rationale Line 87: polished? Is this right? Line 179: extracts As Figure 1A, please show a schematic of a Nanobody; show where HA-tag is located and signal peptide (?). Figure 1: how was this alignment generated? What program? What is the gap at 10 and 72/73? Numbering explanation and IMGT link should go into Methods. Please define CDR and FR. Yellow highlights are hard to see. What are VH hallmarks? VHH and VL need to be explained better. Figure 2A: y-axis units? Figure 2B/C: label the size markers more clearly and increase size of all labels (including B and C). Distinguish in the Figure that B is an SDS-PAGE and C is a Western blot with HA antibody. Figure 3: make Nb labels bigger. The babA Nb used in Figure 3 – is this the same Ctrl Nb used in Figs 6 & 7? If so make labelling consistent. Delta MFI is change in MFI – please define this better. Figure 3C – y-axis is Proliferation score (%) Line 224: a non-targeting.... Table 1: please define the kinetic parameters being measured here? Please include the kinetic parameters and Tm for the control Nb. The lines/data rows don’t align –fix Figure 4 make all y-axis labels, tick labels and legend labels much bigger. Line 239: insert? Line 228 & 331: alamarBlue Table 2 should have Nanobody not Nanobodies. What does melting temperature tell you for Nanobodies? This is not explained. Figure 5: should contain a line crossing the x-axis as to where the Tm is determined, (AU) arbitrary units? Should be defined in legend. Line 267: p. i. is a not widely used acronym – rephrase. Lines 263-271: combine in one paragraph. Figure 6: What happened to Nb22? The red superimposed kidneys and bladder really distract from the data. Please use hollow dashed outline to highlight. Please comment on the conservation between mouse and human CD33 in the ectodomain. Do you expect the nanobodies to cross-react at the same epitopes? Figure 7: %IA/g is not defined anywhere; presentation? A) instines?? x-axis labels are weird. Table 3 is superfluous, put data in Supplementary. Line 291: fix typographical error Line 295: ‘As means to fight these grim odds...’???? Line 340: ‘reunited’...?? and reunites in Abstract Line 25 Line 344: ‘reuniting globally’...?? Line 346/7: ‘concrete representation’....?? Check all references: 22 – incomplete; 29 – what journal; 34 – incomplete; 35 – larger font?; 38 – incomplete, 41 - incompleter
Author Response
Response to the reviewers
Manuscript ID: ijms-642026:
Identification of a lead Nanobody against the acute myeloid leukemia marker CD33
First of all, we would like to thank the reviewers for their positive and constructive feedback and helpful suggestions to improve our paper. The original sentences have been rephrased throughout the text to increase readability, to correct typo’s and/or remove inconsistencies in abbreviations. We amended the text of our manuscript to address all the comments raised by the reviewers in the following manner:
Reviewer #2
The authors have given a reasonable account of the successful generation of CD33 Nanobodies and demonstration of their specificity. This will be of interest for those in the leukaemia field, seeking to develop new therapeutic agents. The authors must improve both the readability and presentation of their manuscript by addressing the comments below:
English language and style: Moderate English changes required
Title: there is no lead Nanobody candidate here. No experimental data presented gives clear data on a ‘lead’ candidate therefore remove from Title and abstract.
Answer from authors to comment: From the in vivo and ex vivo biodistribution data, anti-CD33 Nb_7 showed the best biodistribution profile. Nevertheless, we understand the remark of the reviewer that the data are, at this stage, not yet conclusive to promote Nb_7 as a clear ‘lead’ to develop an independent therapeutic or diagnostic tool. Therefore, we removed ‘lead’ in the title and removed sentences in abstract, intro, and discussion referring to Nb_7 as a ‘lead’ compound.
Check all Figures and legends for consistent usage line 22: tumour-bearing.
Answer from authors to comment: We agree, ‘tumour’ has been changed into ‘tumor’, also ‘labelled’ has been changed into ‘labeled’, etc throughout the text.
437 000 – number looks weird, check other numbers too eg Line 114.
Answer from authors to comment: We agree, the text was changed into “with over 400,000 cases worldwide in 2018”
line 34: a fruit of labor??
Answer from authors to comment: We agree this is verbose. The sentence has been amended into “To date, human leukemia stem cells (LSCs) causing AML are the most and best studied cancer stem cell population”.
Line 52: rationale
Answer from authors to comment: OK, the sentence now starts with: ”The main rationale of using…”
Line 87: polished? Is this right?
Answer from authors to comment: We removed ‘polished’ from the sentence. The sentence was amended and now reads as follows: “Recombinant pMECS plasmids encoding for HA- and His6-tagged Nbs were transformed in Escherichia coli WK6 cells for the periplasmic expression of Nbs. Following extraction of the Nb proteins from the periplasm, they were purified by immobilized metal affinity chromatography (IMAC) and size-exclusion chromatography (SEC), as previously described”
Line 179: extracts
Answer from authors to comment: OK the typo was corrected and now reads: “the periplasmic extracts of 78 clones…”
As Figure 1A, please show a schematic of a Nanobody; show where HA-tag is located and signal peptide (?).
Answer from authors to comment: A schematic representation of the annotated Nb gene construct in the phagemid vector pMECS has been added as new Figure 1A.
Figure 1: how was this alignment generated? What program? What is the gap at 10 and 72/73? Numbering explanation and IMGT link should go into Methods.
Answer from authors to comment: The following sentence has been added to the Methods section: “The inserted DNA sequences were translated into amino acid sequences and categorized in different Nb-families. The Nb sequences were analyzed using the CLC Main Work Bench 8 software (Qiagen, Hilden, Germany), aligned according to the international ImMunoGeneTics database (http://imgt.cines.fr) and manually annotated [23].”
Additionally the following text has been added to the legend of Figure 1 (now figure 1B): “Amino acid sequences of the anti-CD33 Nbs (numbering according to IMGT – the International ImMunoGeneTics information system in http://imgt.cines.fr/) [23]. The CDR1, CDR2 and CDR3 regions are highlighted in cyan, green and purple, respectively. The amino acid sequence of the CDR3 region is displayed in alphabetical order. Position 10 in the Framework region-1 (FR1- IMGT) and position 73 in the FR3-IMGT are gaps introduced to align to other V-GENE groups or subgroups. For Nb_7, Nb_21 and Nb_22 an amino acid deletion relative to other sequences occurred at position 85 (represented by a dash). CDR, complementarity determining region; FR, framework region.”
Please define CDR and FR.
Answer from authors to comment: Please see last sentence in previous answer to reviewer.
Yellow highlights are hard to see.
Answer from authors to comment: The yellow highlighting has been removed.
What are VH hallmarks? VHH and VL need to be explained better
Answer from authors to comment: The following paragraph was added to explain the VHH hallmarks: “Analysis of the amino acid sequence of the Nbs revealed the presence Y (or H) at position 42, and R at position 50 (positions refer to the IMGT amino acid numbering). These hallmark amino acids indicate that the V-domain originated from a heavy chain only antibody [28]. The Nb_12 possesses V42 and P50, these amino acids are encountered in the VH domain of classical hetero-tetrameric antibodies [23,29]. However the presence of E118 and D119 instead of W118 and G119 indicates that this Nb_12 would fail to interact with a VL domain, hence it was also derived from a heavy-chain only antibody [29].”
Figure 2A: y-axis units? Figure 2B/C: label the size markers more clearly and increase size of all labels (including B and C). Distinguish in the Figure that B is an SDS-PAGE and C is a Western blot with HA antibody.
Answer from authors to comment: The figures and legends have been amended as requested.
Figure 3: make Nb labels bigger. The babA Nb used in Figure 3 – is this the same Ctrl Nb used in Figs 6 & 7? If so make labelling consistent.
Answer from authors to comment: Figure 3 has been modified to increase the font of the labelling. The non-targeting ctrl Nb used for the in vivo and ex vivo biodistribution is not the same as the non-targeting control Nb in Figure 3.
The origin of the ctrl Nb used for in vivo imaging is mentioned in Materials and Methods section by citing the original reference ((ref 27). Also in the result section we added: “(the control Nb is an anti-idiotypic described by Lemaire et al. [27] that was selected for its low background in naïve mice)”
Delta MFI is change in MFI – please define this better.
Answer from authors to comment: OK, the text has been amended and MFI has been changed into ΔMFI throughout the text: “Anti-CD33 Nbs were only considered as binders if the difference in mean fluorescence intensity (ΔMFI) signal was at least 3 times higher than the signal obtained with a non-targeting Nb. The ΔMFI obtained…”
Figure 3C – y-axis is Proliferation score (%)
Answer from authors to comment: This has been corrected in amended figure 3C
Line 224: a non-targeting....
Answer from authors to comment: the text was changed and “non-targeting” was used throughout the text.
Table 1: please define the kinetic parameters being measured here? Please include the kinetic parameters and Tm for the control Nb. The lines/data rows don’t align –
Answer from authors to comment: The text and legend of Table 1 have been modified to explain the kinetic association and dissociate parameters of interaction and the Equilibrium dissociation constant. The affinity measurements of the control Nb, Nb_19 have been added in the table and in the legend it is specified that these data are for the interaction with its cognate antigen (“The binding parameters for Nb_19 are for the interaction with its cognate antigen BabA (H. pylori’s adhesion protein).”
Figure 4 make all y axis labels, ticks and legend labels much bigger.
Answer from authors to comment: The labels have been made larger to increase visibility and readability.
Line 239: insert?
Answer from authors to comment: The legend has been amended to remove “insert”.
Line 228 & 331: alamarBlue
Answer from authors to comment: “alamarBlue” has been spelled correctly now.
Table 2 should have Nanobody not Nanobodies.
Answer from authors to comment: OK, was amended in this revised version.
What does melting temperature tell you for Nanobodies? This is not explained.
Answer from authors to comment: Our mistake, we added this explanation now and the paragraph reads: “To examine the thermal stability of anti-CD33 Nbs, we evaluated the melting temperature by using the ThermoFluor®assay (Table 1 and Figure 5). The melting temperature of a protein is a measure of its stability, a parameter considered as an important indicator of good pharmaceutical properties. The melting temperatures of anti-CD33 Nbs ranged from 52.67 ± 0.04 to 67.80 ± 0.08 °C. This is above the temperature of 50°C, which is needed for practical labeling of anti-CD33 Nbs with 99mTc for the in vivo characterization experiments.”
Figure 5: should contain a line crossing the x-axis as to where the Tm is determined, (AU) arbitrary units? Should be defined in legend. The Tm was determined by using a mathematical fit: the Boltzmann Equation.`
Answer from authors to comment: OK, the legend of figure 5 has been completed and reads now: “Figure 5. An example of melting temperature measurement of anti-CD33 Nb_12, as determined by ThermoFluor® assay. AU stands for arbitrary units. The melting temperature for this Nb (58.00 ± 0.23°C) was determined using the Boltzmann’s Equation.”
A crossing line at the x-axis is not necessary as the Tm was determined using the Boltzmann’s Equation.
Line 267: p. i. is a not widely used acronym – rephrase.
Answer from authors to comment: Although “p.i.” was introduced the first time it occurred in the text (M&M), since it was only used 2 more times we replaced p.i. by post injection (everywhere).
Lines 263-271: combine in one paragraph.
Answer from authors to comment: The paragraphs on the in vivo imaging have been combined into one single paragraph.
Figure 6: (1) What happened to Nb22? (2) The red superimposed kidneys and bladder really distract from the data. Please use hollow dashed outline to highlight. (3) Please comment on the conservation between mouse and human CD33 in the ectodomain. Do you expect the nanobodies to cross-react at the same epitopes?
Answer from authors to comment: (1) We agree, we should have explained what happened with Nb_22. Therefore, we now added in the revised version following explanation: “Although Nb_22 had the highest thermostability (Tm 58°C) of the three Nbs of the same family (Tm of Nb_7 = 54°C, and Nb_21 = 53°C), this Nb_22 failed to withstand the labeling conditions. Hence, it was not used for in vivo biodistribution studies. All successfully 99mTc-labeled Nbs showed a radiochemical purity of > 95%. Biodistribution and tumor-targeting of 99mTc-Nbs ….. “
(2) The red colour is the saturated signal. Even if we choose another colour, we will still have the superimposed high signal in Kidney and Bladder. Removign the kidney and bladder signal is manipulating the figure. For sure this is something we cannot do.
(3) The possible cross reactivity of the Nbs on human/mouse CD33: We thank the reviewer for raising this point. Especially for Nb_21 showing high liver and spleen signal this might be relevant. We added following sentences:“Mice injected with 99mTc-Nb_21 showed higher accumulation of radioactivity in liver and spleen compared to the group of mice injected with the other 99mTc-labeled Nbs. This might be due to cross reactivity of Nb_21 with mouse CD33, as mouse and human CD33 share 62% amino acid sequence identity for their ectodomain, but this hypothesis needs to be confirmed by further investigation.”
Figure 7: %IA/g is not defined anywhere; presentation? A) instines?? x-axis labels are weird.
Answer from authors to comment: In Materials and Methods, the %IA/g has been defined under heading “In vivo biodistribution of radiolabeled Nbs”, as “… and expressed as % injected activity per gram (% IA/g) tissue”
The “intestines” typo in x-axis of Figure 7A has been corrected.
Table 3 is superfluous, put data in Supplementary.
Answer from authors to comment: We do not entirely agree that the contents of the table 3 are superfluous, as they provide the real %IA/g tissue, that is lost somehow from the graphical representation (Figure 7). However, we followed the recommendation of the reviewer and put the table as a supplementary material.
Line 291: fix typographical error
Answer from authors to comment: Done
Line 295: ‘As means to fight these grim odds...’????
Answer from authors to comment: We agree with the comment of an inappropriate use of verbose words. The sentences were amended and now read: “To improve these numbers, antibody-based therapies have been pursued since the late 80-ies. However, they all showed very limited success due to low efficacy, toxicity and immunogenicity issues [10].”
Line 340: ‘reunited’...?? and reunites in Abstract Line 25
Answer from authors to comment: Reunites was substituted in Abstract by “exhibits”. The other sentence has been removed as the whole discussion was largely rewritten.
Line 344: ‘reuniting globally’...??
Answer from authors to comment: Also this sentence in the discussion has been deleted.
Line 346/7: ‘concrete representation’....??
Answer from authors to comment: Sentence was amended to avoid these words.
Check all references: 22 – incomplete; 29 – what journal; 34 – incomplete; 35 – larger font?; 38 – incomplete, 41 - incompleter
Answer from authors to comment: The bibliographic list has been carefully checked to avoid incomplete references. Only old reference 34 that became in the revised text reference 43 still doesn’t contain a journal name as we refer to a patent. The citation (patent application has been added).
Reviewer 3 Report
This paper by Romao et al is a solid piece of work describing generation and characterisation of novel Nanobodies specific to CD33 in vitro and in vivo. A lot of work went into this work. The nanobodies were produced via a well-established method (phage display) and recombinant expressed in E.coli followed by in vitro experiments and testing in vivo xenograft animal model. I’m happy to recommend this paper for publication after a few minor and major adjustments of the discussion have been taken into consideration.
Paper is well-written, with a few grammatical errors and clunky sentence structures, mild editing required. Please update the literature Do not use the term “sacrifice” when referring to the euthanasia of animals. Accurate terminology such as “euthanize” or “kill” must be used in all instances. Would be good to have a radiochemical stability assay, as this is not shown/referred to at all Figure 2 A shows a single SEC profile but seems to refer to all the nbs, please adjust the legend. What happened to Nb22? I haven’t seen any in vivo data of Nb22. More detail as to why the CD33 Nbs lacked any leukemogenicity activity There is no measure of the serum stability of the labeled Nb – would add value to the manuscript. PET scans were done 1 h post injection – please explain why this time point was chosen? An in vivo blocking/competitive study should be done to confirm specificity. What is the blood clearance of the Nb? A blood half-life curve over at least 2-3 hours would bring a lot of value to the paper. The first 30 min would be really interesting to investigate. A key concern involved the fact that the discussion is written as a summary instead of a proper discussion and lacks of comparison to the literature nanobodies do not have an anti-proliferative effect on the THP-1 AML cancer cell line / the CD33 Nbs could be used as imaging agents for the diagnosis of AML (pro/cons0 – especially please discuss these two points in detail and compare to the literature.Author Response
Response to the reviewers
Manuscript ID: ijms-642026:
Identification of a lead Nanobody against the acute myeloid leukemia marker CD33
First of all, we would like to thank the reviewers for their positive and constructive feedback and helpful suggestions to improve our paper. The original sentences have been rephrased throughout the text to increase readability, to correct typo’s and/or remove inconsistencies in abbreviations. We amended the text of our manuscript to address all the comments raised by the reviewers in the following manner:
Reviewer #3
This paper by Romao et al is a solid piece of work describing generation and characterisation of novel Nanobodies specific to CD33 in vitro and in vivo. A lot of work went into this work. The nanobodies were produced via a well-established method (phage display) and recombinant expressed in E.coli followed by in vitro experiments and testing in vivo xenograft animal model. I’m happy to recommend this paper for publication after a few minor and major adjustments of the discussion have been taken into consideration.
Comments & suggestions:
Paper is well-written, with a few grammatical errors and clunky sentence structures, mild editing required.
Answer from authors to comment: We would like to thank the reviewer for the supportive comment. We removed several grammatical errors and many sentences/paragraphs were rewritten to avoid ‘clunky sentences’.
Please update the literature
Answer from authors to comment: Several new citations were introduced.
Do not use the term “sacrifice” when referring to the euthanasia of animals. Accurate terminology such as “euthanize” or “kill” must be used in all instances.
Answer from authors to comment: “Sacrificed” has been substituted by “killed”.
Would be good to have a radiochemical stability assay, as this is not shown/referred to at all Figure 2 A shows a single SEC profile but seems to refer to all the Nbs, please adjust the legend.
Answer from authors to comment: A radiochemical stability of radiolabeled Nbs has not been performed. However, the radiochemical purity of radiolabeled Nbs was performed by iTLC and showed to be higher than 95%. Based on vivo biodistribution data, we expect that injected radiolabeled Nbs show a good stability in vivo (for the short residence time). Indeed, except Nb_21, other Nbs show relatively low background signal, except kidneys and bladder (the clearance route of unbound fractions). In case of in vivo low stability, a higher background signal and no tumor targeting should be observed. With a Tm of >50°C, it is well accepted that the protein tolerates exposure to body temperatures (37°C)for prolonged times. For sure longer than 2 hours, the animals were killed for ex vivo biodistribution measurements, so we think we are safe in our analysis.
The legend of Figure 2 has been modified: “(A) Size exclusion chromatography (SEC) profile of the Nb_12 showing a single peak of protein. (All other Nbs gave comparable chromatograms).”
What happened to Nb22? I haven’t seen any in vivo data of Nb22.
Answer from authors to comment: Entirely justified comment. Was also remarked by reviewer #2. Thus, we should have explained what happened with Nb_22. Therefore, we now added in the revised version following explanation: “Although Nb_22 had the highest thermostability (Tm 58°C) of the three Nbs of the same family (Tm of Nb_7 = 54°C, and Nb_21 = 53°C), this Nb_22 failed to withstand the labeling conditions. Hence, it was not used for in vivo biodistribution studies. All successfully 99mTc-labeled Nbs showed a radiochemical purity of > 95%. Biodistribution and tumor-targeting of 99mTc-Nbs ….. “
More detail as to why the CD33 Nbs lacked any leukemogenicity activity
Answer from authors to comment: We agree that this observation needs some additional explanation. In the result section we wrote: “The anti-proliferative signal obtained after 4-hour incubation of THP-1 cells with 5 µg of each anti-CD33 Nb was the same as the signal obtained from the non-targeting Nb, indicating the absence of an anti-proliferative activity of the Nbs under our experimental conditions.” The text (underlined here) is important as it might be that under modified conditions an anti-proliferating activity might be revealed. In the discussion (nearly half of it) we describe in detail that this should be obtained by generating biparatopic, bivalent Nb constructs of by introducing antibody drug conjugates.
There is no measure of the serum stability of the labeled Nb – would add value to the manuscript.
Answer from authors to comment: We understand the concern of the reviewer. The serum stability of 99mTc-labeled Nbs was not performed. We think that serum stability is an important factor which can influence the in vivo biodistribution profile of radiolabeled compounds, resulting in a high background signal or loss of targeting capacity. In this case, except Nb_21, other Nbs show a typical Nb-profile, suggesting that radiolabeled Nbs are stable in vivo. The serum stability over time is much more important for larger radiolabeled proteins, with prolonged circulation time. Nbs are characterized by a very fast targeting and fast blood clearance. (see also response to blood clearance, below)
PET scans were done 1 h post injection – please explain why this time point was chosen?
Answer from authors to comment: Based on Our SPECT scans of mice are performed 1-hour post-injection, based on empirical evidence and standard protocols used for radiolabeled Nbs.
An in vivo blocking/competitive study should be done to confirm specificity.
Answer from authors to comment: This is an obvious suggestion to demonstrate specificity. However, to determine the in vivo specificity of our radiolabeled anti-CD33 Nbs, we have used a radiolabeled non-targeting Nb. In various studies, we have used this strategy to confirm the target-specificity of radiolabeled Nbs. In vivo blocking study can indeed also be performed as an extra condition to confirm the tumor-specificity of radiolabeled anti-CD33 Nbs. However, in order to reduce the number of animals (3Rs), we haven’t included this extra condition.
What is the blood clearance of the Nb? A blood half-life curve over at least 2-3 hours would bring a lot of value to the paper. The first 30 min would be really interesting to investigate.
Answer from authors to comment: This hasn’t been formally investigated for the CD33 Nbs. However, from previous studies (and all Nbs behave similarly as they share the same MW and dimensions) it is well accepted that: “Calculated blood half-lives of the initial phase are situated around 1 min and those of the slow phase around 30 min. At 1 h p.i., the percentage of injected activity per total blood volume generally decreases below 0.5.” (copied form D’Huyvetter et al. Expert Opinion Drug Delivery (2014) Volume 11 issue 12)
A key concern involved the fact that the discussion is written as a summary instead of a proper discussion and lacks of comparison to the literature nanobodies do not have an anti-proliferative effect on the THP-1 AML cancer cell line / the CD33 Nbs could be used as imaging agents for the diagnosis of AML (pro/cons0 – especially please discuss these two points in detail and compare to the literature.
Answer from authors to comment: We understand the critic of the reviewer. The summary of the results section has been cut to a minimum. Furthermore, the remaining part of the discussion was entirely rewritten to meet his/her critic. We now focus more on the absence of the anti-proliferative effect on AML cells and how to remediate this by generating biparatopic, bivalent Nbs or ADC or radio-nuclide conjugated Nbs.
Round 2
Reviewer 2 Report
Additional points identified in the second round of review, identified below, need to be addressed before this manuscript is considered further.
Figure 1A and B is now confusing and hard to understand. The colour schemes should be more carefully chosen here. The Nanobody sequence portion should be white/grey and CDR1, 2 and 3 with colours depicted in this domain. The same green and magenta used for other parts (AAA linker and HA-tag) are confusing, please use different colours. Lines/dashed lines should extend and expand from the Nanobody sequence region to the sequence in Figure 1B to show where this sequence is in Fig 1A. QVQL....VTVSS can then be removed from 1A. Remove the weblink from the Fig 1B legend as it is in Methods. Why is the amino acid sequence of the CDR3 region alphabetical – this makes no sense? Or is not explained why? Move CDR definition at the end of legend to where CDR is first mentioned. FR is redundant as this is earlier defined. All Figure labels with the Nbs should be consistent with the text eg use Nb_7 or Nb 7 or Nb7 but be consistent with usage throughout the manuscript The non-targeting Nb labelled consistently on the Figures. Is it the Baba? And is this also known as Nb-19 referred to in Table 1? Abstract: clear rapidly not clear fast The abstract mentions the kidney distribution of the Nbs – but bladder should also be mentioned. Please revise this in the abstract and body of the text (page 10) – “except in the kidneys and bladder”. Please provide some explanation for this ie impact of glomerular filtration, binding CD33 antigen lodged in the kidney. Please provide more details on the biotinylated CD33 recombinant protein including catalog number and co-ordinates (in amino acids) of the ectodomain. Alamarblue is still mis-spelt Alarmarblue a few times throughout the manuscript. Starting off with ‘After 4 rounds of phage selection...’ in Section 3.1 (Results) gives no context of the problem/question being addressed. Please provide in Section 3.1 a one paragraph narrative on how the project was initiated. Further to Section 3.1 – describing the Nbs were divided into 10 different families based on amino acid identity is a bit disingenuous. If you were characterising dozens of Nbs this would be useful. Remove this and state that “while all Nb V domains were different, Nb7, Nb21 and Nb22 exhibited the most amino acid similarity”. Also revise Discussion where families are mentioned. Remove the word anti-proliferative from the manuscript. In the Figure legend “The impact of Nbs on THP-1 proliferation was determined....’’. In the text “THP-1 cells proliferative capacity was not impacted by the CD33 Nbs compared to control...” The sentence at the bottom of Table 2 should be moved to the Table 2 caption. Figure 5: what are the temperature increments at which the ThermoFluor melt analysis was performed at? This should be stated in Methods and in the Figure legend. Provide some discussion about whether nanobodies can cross blood-brain barrier. Reference 46 is incomplete. Figure 6: This is an in vivo biodistribution study not ex vivo. BiTE in Discussion is not defined.
Author Response
Reviewer #2:
Figure 1A and B is now confusing and hard to understand. The colour schemes should be more carefully chosen here. The Nanobody sequence portion should be white/grey and CDR1, 2 and 3 with colours depicted in this domain. The same green and magenta used for other parts (AAA linker and HA-tag) are confusing, please use different colours. Lines/dashed lines should extend and expand from the Nanobody sequence region to the sequence in Figure 1B to show where this sequence is in Fig 1A. QVQL....VTVSS can then be removed from 1A.
Answer from authors to comment:
The figure has been redesigned and the colour scheme adjusted as suggested by the reveiwer.
Remove the weblink from the Fig 1B legend as it is in Methods.
Answer from authors to comment:
We removed the weblink from the legend of Figure 1B. The legend is modified: “…Amino acid sequences of the anti-CD33 Nbs (numbering according to IMGT – the International ImMunoGeneTics information system)” + citation article 23
Why is the amino acid sequence of the CDR3 region alphabetical – this makes no sense? Or is not explained why?
Answer from authors to comment:
The drawback of Nbs is that their genes can easily be synthesized when the amino acid sequence is available. This is done by companies (Creative biologics), which commercialise these Nbs without giving any reference to those that generated and characterized the Nbs. Also, competitive academic research groups synthesize our characterized Nbs and then make them available for others and receive all credits for it. To stop this corruption, we prefer to give the amino acid sequence with the CDR3 in alphabetic order. So, the reader has access to the length and amino acid contents of the CDR3 and the whole Nb, but it becomes impossible to synthesise the gene. If readers want to use our Nbs they should contact us and we are willing to share the sequence and to collaborate. In the legend we now wrote: “The amino acid sequence of the CDR3 region is displayed in alphabetical order, its exact sequence can be obtained from the corresponding author”
Move CDR definition at the end of legend to where CDR is first mentioned. FR is redundant as this is earlier defined.
Answer from authors to comment:
In the 3.1 section, line 4 has been modified: “…the amino acid sequence from the complementarity determining region (CDR) 3…”
We removed the CDR and FR definition from the legend of Figure 1.
All Figure labels with the Nbs should be consistent with the text e.g. use Nb_7 or Nb 7 or Nb7 but be consistent with usage throughout the manuscript.
Answer from authors to comment:
In the whole manuscript we use now “Nb_X”.
The non-targeting Nb labelled consistently on the Figures. Is it the Baba? And is this also known as Nb-19 referred to in Table 1?
Answer from authors to comment:
The non targeting control Nb for the in vitro work was a Nb against a bacterial adhesin. This is clearly stated in M&M: “Besides anti-CD33 Nbs, a non-targeting control Nbs was used for in vitro binding and specificity experiments. This was Nb-ER19 from Moonens et al. [24], a Nb directed against a bacterial adhesion protein of Helicobater pylori (BabA), which is referred to herein as aBabA Nb_19.” and in the legend of Figure 3B and 3C. It is also mentioned in table 1 (this was added because one of the reviewers asked for it, although we are not sure of its relevance.
In contrast, the non-targeting control Nb used for the in vivo and ex vivo biodistribution is a different Nb. The origin of this control Nb (to which we refer to as ctrl_Nb)is clearly indicated in the M&M section: “For the in vivo binding studies we preferred to use the anti-mouse idiotype Nb_R3B23 characterized by Lemaire et al. [27] as non-targeting control (here denoted as ctrl_Nb)”.
Also in the result section we explain the reason to use this Nb added: “(the control Nb is an anti-idiotypic described by Lemaire et al. [27] that was selected for its low background in naïve mice)”
Abstract: clear rapidly not clear fast
Answer from authors to comment:
It has been modified to “…tissue penetration and to clear rapidly from blood…”
The abstract mentions the kidney distribution of the Nbs – but bladder should also be mentioned. Please revise this in the abstract and body of the text (page 10) – “except in the kidneys and bladder”.
Answer from authors to comment:
The abstract has been modified to “…with low background signal, except in the kidneys and bladder…”.
The text on page 10 has also been modified to “…with low background signal, except kidneys and bladder.”
Please provide some explanation for this i.e. impact of glomerular filtration, binding CD33 antigen lodged in the kidney.
Answer from authors to comment:
In Discussion, this has been added or modified: “However, kidneys will be the main dose-limiting organ. Indeed, the rapid renal clearance of the unbound labeled Nb-fraction will result in a decreased radiation of other non-target organs, but it may lead to radiation-induced nephropathy. Different strategies can be assessed to reduce the renal accumulation of radiolabeled Nbs, such as the removal of the His6-tag, co-injection of positively-charged amino-acids or Gelofusin… …SPECT/CT imaging at 1-hour post-injection showed high non-specific accumulation in kidney and bladder for all 99mTc-radiolabeled Nbs. The high kidney and bladder accumulation of small hydrophilic compounds is a well-known phenomenon, which occurs as a result of a rapid renal clearance of the unbound fraction… “
Please provide more details on the biotinylated CD33 recombinant protein including catalog number and co-ordinates (in amino acids) of the ectodomain.
Answer from authors to comment:
The required information has been added.
In section 2.5 “recombinant CD33 ectodomain (Acro Biosystems, CD3-H5226, Asp 18 – His 259)” has been added. In section 2.5 “biotinylated CD33 recombinant protein (Acro Biosystems, CD3-H82E7, Asp 18 - His 259)” has been added.
Alamarblue is still mis-spelt Alarmarblue a few times throughout the manuscript.
Answer from authors to comment:
Our mistake, this has been corrected now.
Starting off with ‘After 4 rounds of phage selection...’ in Section 3.1 (Results) gives no context of the problem/question being addressed. Please provide in Section 3.1 a one paragraph narrative on how the project was initiated.
Answer from authors to comment:
We agree. However, details of immunising a llama and generating the Nb library in phage display vectors was defined in M&M. In the result section we now start with: “A Nb gene library, in pMECS phage display vector [22], of 108 transformants was constructed from peripheral blood of a llama immunized with recombinant CD33” before continuing with “After 4 rounds of phage display selection on recombinant CD33 coated on microtiter plates, the periplasmic extracts of 78 individual clones….”
Further to Section 3.1 – describing the Nbs were divided into 10 different families based on amino acid identity is a bit disingenuous. If you were characterising dozens of Nbs this would be useful. Remove this and state that “while all Nb V domains were different, Nb7, Nb21 and Nb22 exhibited the most amino acid similarity”.
Answer from authors to comment:
We respectfully disagree with the reviewer. The analysis of the amino acid sequence and the CDR3 in particular is relevant as it unveils which Nbs will bind to an identical epitope. The presence of key amino acids in FR2 and the length of the CDR3 is suggestive (for specialists) of how the antigen binding loops might be organised. However, we do agree that framing was missing. We therefore, added in the result section: “One of the Nb-families contained three representatives, namely Nb_7, Nb_21 and Nb_22, with up to five point mutations. These three Nbs belonging the same family are derived from a single B cell lineage (sharing the same VDJ rearrangement). Nbs from the same family will recognize an identical epitope, however, the affinity for the antigen can be different due to somatic hypermutations that accumulated as a result of the affinity maturation process during immunization of the llama.”
Also revise Discussion where families are mentioned.
Answer from authors to comment
In the discussion we further clarify: “…. Nb_7, Nb_21 and Nb_22 showed significant homology in their CDR3 region, and minor amino acid variations in the remaining sequence. These clones are derived from the same V-D-J recombination event, revealing their common B cell origin. Although the somatic hypermutation (or PCR errors) might have affected the affinity for the antigen, the dominant importance of the CDR3 in recognizing the epitope, makes that they will bind to the same epitope (ref), a finding that was confirmed here from our epitope binning using SPR.”
Remove the word anti-proliferative from the manuscript.
Answer from authors to comment:
In the result section ‘anti-proliferative’ was changed and the sentence now reads: “The impact of the six Nbs recognizing CD33 on the proliferation of THP-1 cells was evaluated via the alamarBlue assay (Figure 3C).… …. indicating that the proliferative capacity of THP-1 cells was not impacted by the anti-CD33 Nbs under our experimental conditions”
In the legend of the Figure 3 and the discussion, this has been modified into: “To determine the in vitro therapeutic potential of selected anti-CD33 Nbs, we evaluated the impact of anti-CD33 Nbs on the proliferation of THP-1 cells by performing the alamarBlue assay. …”
In the discussion we wrote: “To determine the in vitro therapeutic potential of selected anti-CD33 Nbs, we evaluated the impact of anti-CD33 Nbs on the proliferation of THP-1 cells by performing the alamarBlue assay.”
The sentence at the bottom of Table 2 should be moved to the Table 2 caption.
Answer from authors to comment:
The sentence has been moved.
Figure 5: what are the temperature increments at which the ThermoFluor melt analysis was performed at? This should be stated in Methods and in the Figure legend.
Answer from authors to comment:
We agree. The section “2.6 Thermostability” has been modified as follows: “The melting temperature (Tm) of the anti-CD33 Nbs was determined via a ThermoFluor® assay using CFX ConnectTM Real-Time PCR (Biorad, Pleasanton, USA) and the fluoroprobe Sypro® Orange dye (Molecular Probes, Oregon, USA), as previously described [25]. For the thermal denaturation the samples were heated from 10°C to 95°C with stepwise increments of 0.5°C per 30 secondsand a 10 second hold step every point, followed by the fluorescence reading. The melting temperatures were calculated using the Boltzmann equation.”.
To the legend of Figure 5 the following sentence has been added: “The samples were heated from 10°C to 95°C with stepwise increments of 0.5°C per 30 seconds.”
Provide some discussion about whether nanobodies can cross blood-brain barrier.
Answer from authors to comment:
The possible crossing of the BBB by Nbs is controversial. While some investigators indicated that Nbs could pass passively through the BBB, the data from others demonstrated that the amount of Nbs passing through the BBB remains far below a relevant therapeutic level. Therefore, we prefer not to speculate on this issue, as we haven’t explored the amount of labelled Nb in the brain.
Reference 46 is incomplete
Answer from authors to comment:
The reference is now complete.
Figure 6: This is an in vivo biodistribution study not ex vivo.
Answer from authors to comment:
Indeed, Figure 6 represents results obtained from an in vivo study. The paragraph under Figure 6, starting with “The ex vivo biodsitribution study…” refers to figure 7A & B, and Table 3, representing results from the ex vivo study.
BiTE in Discussion is not defined.
Answer from authors to comment:
Our mistake, this is added now: “…easily allow for constructing a Bi-specific T-cell engagers (BiTE)-like format…”
Reviewer 3 Report
The authors did well by addressing the comments. However there is still a bit of work to be done with regards to the discussion. The discussion improved but it is difficult to read. The telling style should be held simply but most importantly it should be read fluently.
Author Response
Manuscript ID: ijms-642026:
Identification of a lead Nanobody against the acute myeloid leukemia marker CD33
Response to the reviewers
Reviewer #3:
The authors did well by addressing the comments. However, there is still a bit of work to be done with regards to the discussion. The discussion improved but it is difficult to read. The telling style should be held simply but most importantly it should be read fluently.
Answer from authors to comment:
We are not very sure were we should amend the Discussion. However, we have tried to improve the readability by adding a few words, sentences throughout the discussion.
A conclusion section was also added.